# The transcription factor LAG-1/CSL plays a Notch-independent role in controlling terminal differentiation, fate maintenance, and plasticity of serotonergic chemosensory neurons

Miren Maicas[1], Ángela Jimeno-Martín[1¤], Andrea Millán-Trejo[1], Mark J. Alkema[2], Nuria Flames[1] *

1 Developmental Neurobiology Unit, Instituto de Biomedicina de Valencia IBV-CSIC, Valencia, Spain,
2 Department of Neurobiology, University of Massachusetts Medical School, Worcester, Massachusetts, United States America

¤ Current address: San Jorge University, Zaragoza, Spain
* nflames@ibv.csic.es

## Abstract

During development, signal-regulated transcription factors (TFs) act as basal repressors and upon signalling through morphogens or cell-to-cell signalling shift to activators, mediating precise and transient responses. Conversely, at the final steps of neuron specification, terminal selector TFs directly initiate and maintain neuron-type specific gene expression through enduring functions as activators. *C. elegans* contains 3 types of serotonin synthesising neurons that share the expression of the serotonin biosynthesis pathway genes but not of other effector genes. Here, we find an unconventional role for LAG-1, the signal-regulated TF mediator of the Notch pathway, as terminal selector for the ADF serotonergic chemosensory neuron, but not for other serotonergic neuron types. Regulatory regions of ADF effector genes contain functional LAG-1 binding sites that mediate activation but not basal repression. *lag-1* mutants show broad defects in ADF effector genes activation, and LAG-1 is required to maintain ADF cell fate and functions throughout life. Unexpectedly, contrary to reported basal repression state for LAG-1 prior to Notch receptor activation, gene expression activation in the ADF neuron by LAG-1 does not require Notch signalling, demonstrating a default activator state for LAG-1 independent of Notch. We hypothesise that the enduring activity of terminal selectors on target genes required uncoupling LAG-1 activating role from receiving the transient Notch signalling.

## Introduction

The generation of the plethora of neuron types present in the nervous system is a complex and protracted process that is currently not well understood. In the earliest steps of neuron

**Data Availability Statement:** All relevant data are within the paper and its Supporting information files.

**Funding:** NF research is funded by European Research Council (ERC StG 281920; ERC COG 101002203), Spanish Government (SAF2017-84790-R) and Generalitat Valenciana (PROMETEO/2018/055). A.M.T holds the PRE2018-086632 fellowship from Spanish Government and A.J the ACIF/2015/398 fellowship from Generalitat Valenciana. The funders had no role in study design, data collection and analysis, decision to publish, or preparation of the manuscript.

**Competing interests:** The authors have declared that no competing interests exist.

**Abbreviations:** 5HT, serotonin; bHLH, Basic Helix Loop Helix; CRM, *cis*-regulatory module; CSL, CBF-1, Su (H), LAG-1; ETS, Erythroblast Transformation Specific; HD, Homeodomain; L1, first larval stage; NHR, Nuclear Hormone Receptor; NICD, Notch intracellular domain; sc-RNAseq, single-cell RNAseq; SOX, SRY-related HMG-box genes; TF, transcription factor; TIR-1, Toll and Interleukin 1 Receptor; ZF, Zinc Finger.

specification, morphogens and cell surface signalling molecules specify naive cells towards restricted neuronal progenitors [1]. Sharp differences in gene expression levels between different types of neuronal progenitors are essential for precise neuronal induction. Thus, these signalling molecules act through specific transcription factors (TFs) called signal-regulated TFs. During basal conditions, signal-regulated TFs function as repressors and they shift to temporally and spatially restricted gene activation when the signalling molecule is received [2]. Different mechanisms, including protein instability, posttranslational modifications, negative feedback loops, or cross-inhibition, ensure the transient activating effects of signal-regulated TFs [2,3].

Later in development, during neuronal terminal differentiation, in contrast to the transient activation effects mediated by signal-regulated TFs, the enduring actions of specific combinations of TFs, termed terminal selectors, ensure the direct and sustained transcriptional activation of neuron-type specific effector genes, such as ion channels, neurotransmitter receptors, neurotransmitter biosynthesis enzymes, etc. [4]. Effector genes are ultimately responsible for the specific properties and physiological functions of the neuron, and their expression is sustained throughout the life of the cell. Accordingly, terminal selector TFs are also continuously expressed, and they are necessary not only to establish but also to maintain correct effector gene expression [5,6].

Neuronal terminal selectors have been described in different metazoan groups including mammals [5,7], although they have been best characterised in *C. elegans*, where at least a terminal selector is known for 76 out of the 118 different neuronal types [8]. Interestingly, despite numerous examples of TFs acting as terminal selectors, only a small number of TF families are known to act as neuronal terminal selectors from over 50 different families that are present in the *C. elegans* genome. This includes members of the Erythroblast Transformation Specific (ETS), SRY-related HMG-box genes (SOX), Nuclear Hormone Receptors (NHRs), Basic Helix Loop Helix (bHLH), Zinc Finger (ZF), and, most prominently, the Homeodomain (HD) family [8]. Yet, for many neuronal types in *C. elegans*, not a single terminal selector has been identified so far, thus additional TF families could also be directly involved in neuronal terminal differentiation.

Serotonin (5HT) is a phylogenetically conserved neurotransmitter present in all animal groups. The same enzymes and transporters are used in different organisms to synthesise, release, and reuptake 5HT, which are collectively known as the 5HT pathway effector genes (Fig 1A). *C. elegans* contains 3 different types of 5HT synthesising neurons: the ADF chemosensory neuron, the NSM neurosecretory neuron, and the HSN motorneuron (Fig 1B). Despite sharing the expression of the 5HT pathway genes, each 5HT neuron type expresses a very different set of additional effector genes [9]. Accordingly, the combination of terminal selectors acting in NSM and HSN are different [9,10]. In contrast to the well-characterised NSM and HSN terminal differentiation programs, no terminal selector has been identified for the ADF chemosensory neuron.

Here, we demonstrate that LAG-1, a signal-regulated TF from the CBF-1, Su (H), LAG-1 (CSL) family, which conventionally shows repressor activity until Notch is activated or independent of Notch, is a terminal selector in the ADF neuron acting as an activator independent of Notch signalling. Through *cis*-regulatory analysis, we identify functional CSL binding sites in ADF-active regulatory sequences for all 5HT pathway genes. Importantly, in contrast to previous reported basal repressive roles for LAG-1, these sites are required for ADF effector gene activation but do not mediate basal repression. We find *lag-1* mutants show broad defects in ADF terminal differentiation. LAG-1 is continuously required postembryonically to maintain correct ADF effector gene expression and functions. Finally, we find that LAG-1 ectopic expression is sufficient, in some contexts, to drive ADF effector gene expression. Altogether,

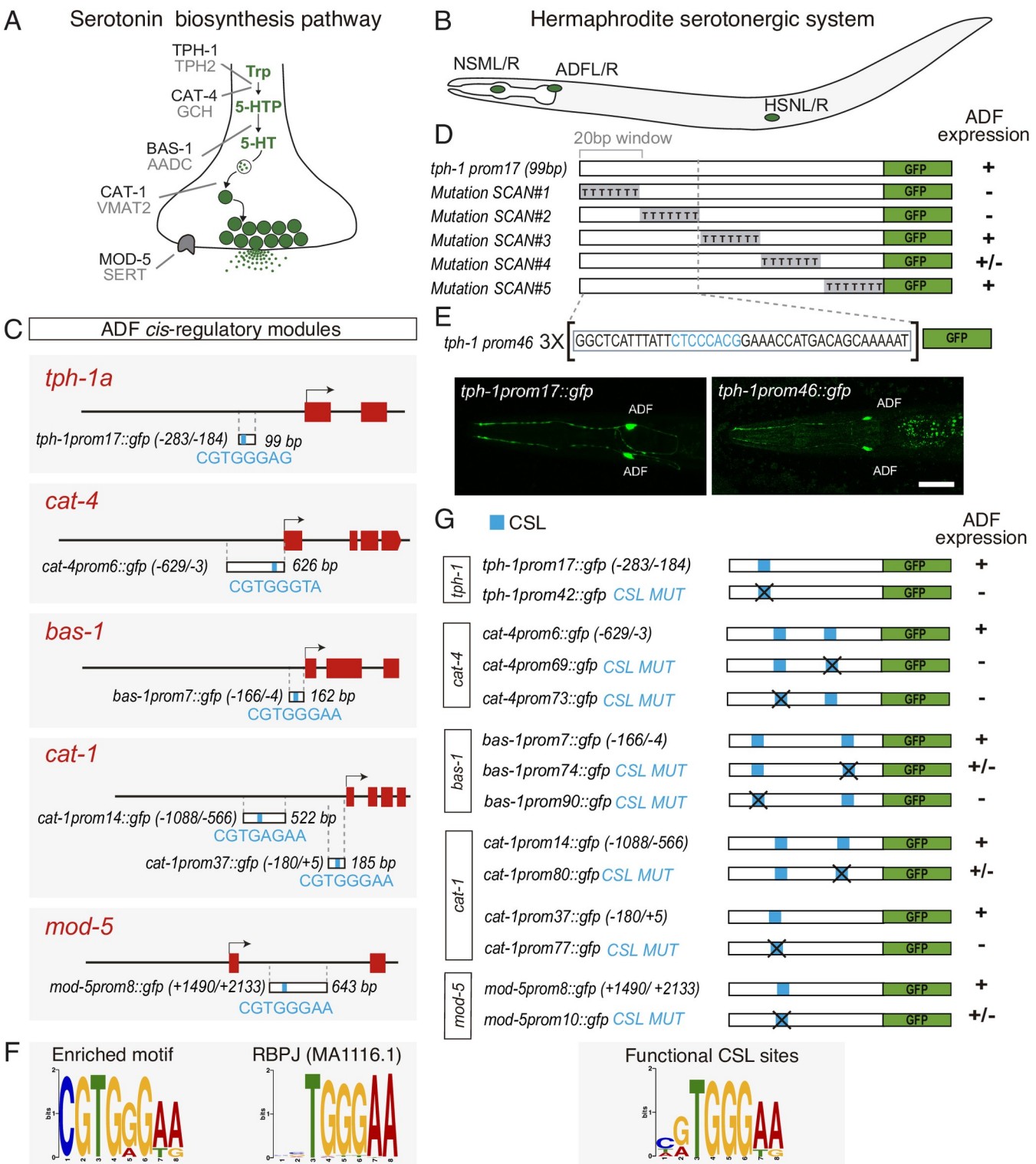

**Fig 1. CSL/LAG-1 binding sites are required for serotonin pathway gene expression in the ADF chemosensory neuron.** (A) Phylogenetically conserved serotonin biosynthetic pathway. *C. elegans* protein names appear in dark font, mammalian in grey. (B) *C. elegans* hermaphrodite serotonin synthesising system is composed of 3 types of bilateral neurons (NSM, ADF, and HSN, L: left, R: right). (C) 5HT pathway gene minimal CRMs active in ADF neurons. The most significant overrepresented motif shared by all CRMs is depicted in blue. Numbers in brackets represent the coordinates of each construct referred to the translational start site. Red boxes represent coding exons. (D) Scanning mutation analysis of *tph-1* minimal CRM unravels the functionality of the first 40 bp. Sequence windows of 20 bp were mutated to poliT. +: >80%; +/−: 80%–20%; −: <20% of mean wild type expression values. See S1 Fig for raw data. (E) Young

adult images showing *tph-1* minimal CRM *tph-1prom17* and the synthetic construct *tph-1prom46* (3X 40 bp window), which are expressed exclusively in the ADF neuron. Scale bar: 10 μm. (F) PWM of the most significant enriched motif and its closest match that corresponds to mammalian RBPJ. (G) Mutational analysis of predicted CSL binding sites. Note that for *cat-1prom14*, in addition to the motif identified in Fig 1C (CGTGAGAA), an extra motif with a better match for CSL consensus site (TATGGGAA) is found. This better match was selected for mutational analysis. +, +/−, and − range values are the same as in D. See S1 Fig for point mutation description and raw data. AADC, aromatic L-amino acid decarboxylase; CRM, *cis*-regulatory module; CSL, CBF-1, Su (H), LAG-1; GCH, GTP cyclohydrolase; PWM, position weight matrix; SERT, serotonin transporter; TPH, tryptophan hydroxylase; Trp, tryptophan; VMAT, vesicular monoamine transporter; 5HT, serotonin; 5HTP, 5-hydroxytryptophan.

these results define LAG-1 as ADF terminal selector. Unexpectedly, we find LAG-1 fulfils its activating role as terminal selector independent from the Notch signalling, thus acting as a constitutive activator. In addition, we also explore LAG-1 actions on *tph-1*/tryptophan hydroxylase gene expression plasticity in the ADF. Our results unravel a complex scenario in which different environmental stimuli modulate *tph-1*/tryptophan hydroxylase gene expression through different TFs and different *cis*-regulatory regions, and some, but not all, of the responses to these stimuli are dependent on LAG-1 function.

## Results

### CSL/LAG-1 binding sites are required for serotonin pathway gene expression in the ADF chemosensory neuron

*Cis*-regulatory analysis of neuron type–specific effector genes has been instrumental to identify the transcriptional regulatory logic directing neuron type differentiation [11]. Thus, we isolated the minimal *cis*-regulatory modules (CRMs) that drive serotonin pathway gene transcription in the ADF neuron (Fig 1C and S1 Fig) [9]. We next focused on the smallest CRM, a 99-bp region located upstream the *tph-1*/tryptophan hydroxylase locus (*tph-1prom17*); this construct drives *gfp* expression exclusively in the ADF neuron (Fig 1C and 1E). We performed serial scanning mutagenesis and identified a 40-bp region inside *tph-1prom-17* that is required for ADF neuron expression (Fig 1D). A synthetic construct with 3 copies of this 40-bp minimal region *(tph-1prom46)* is sufficient to drive *gfp* expression exclusively in the ADF neuron (Fig 1E), while a single copy of the 40-bp region placed in front of a neuronal minimal promoter not expressed in the ADF is unable to drive ADF expression (S1 Fig). Altogether, these data show that *cis*-regulatory information located in this small 40-bp region is necessary and sufficient, when multimerized, to drive gene expression specifically in the ADF neuron.

Terminal selectors directly coregulate expression of neuron type–specific effector genes; thus, we performed de novo motif discovery analysis [12] using the 6 minimal 5HT pathway gene CRMs, including the small 40-bp region in the *tph-1* locus. The most significantly enriched motif highly resembles the RBPJ binding motif (Fig 1C and 1F). RBPJ, also known as CBF1, is the vertebrate member of the highly conserved CSL TF family, which in *C. elegans* is composed by the unique member LAG-1. Disruption of the CSL motif in 5HT pathway gene CRMs leads to strong GFP expression defects in the ADF neuron with no significant up-regulation or ectopic GFP expression observed upon mutation (Fig 1G and S1 Fig), suggesting that LAG-1, through its binding to the CSL motif, could directly activate ADF terminal differentiation.

### LAG-1 controls ADF neuron terminal differentiation

We next analysed ADF neuron terminal differentiation in *lag-1* mutants. LAG-1 is the signal-regulated TF of the Notch signalling pathway that mediates cell–cell interactions during development. *lag-1(q385)* allele is a premature STOP (Fig 2A) that displays the strongest phenotypes of all characterised *lag-1* alleles, including lethality, and it is considered to be a null allele

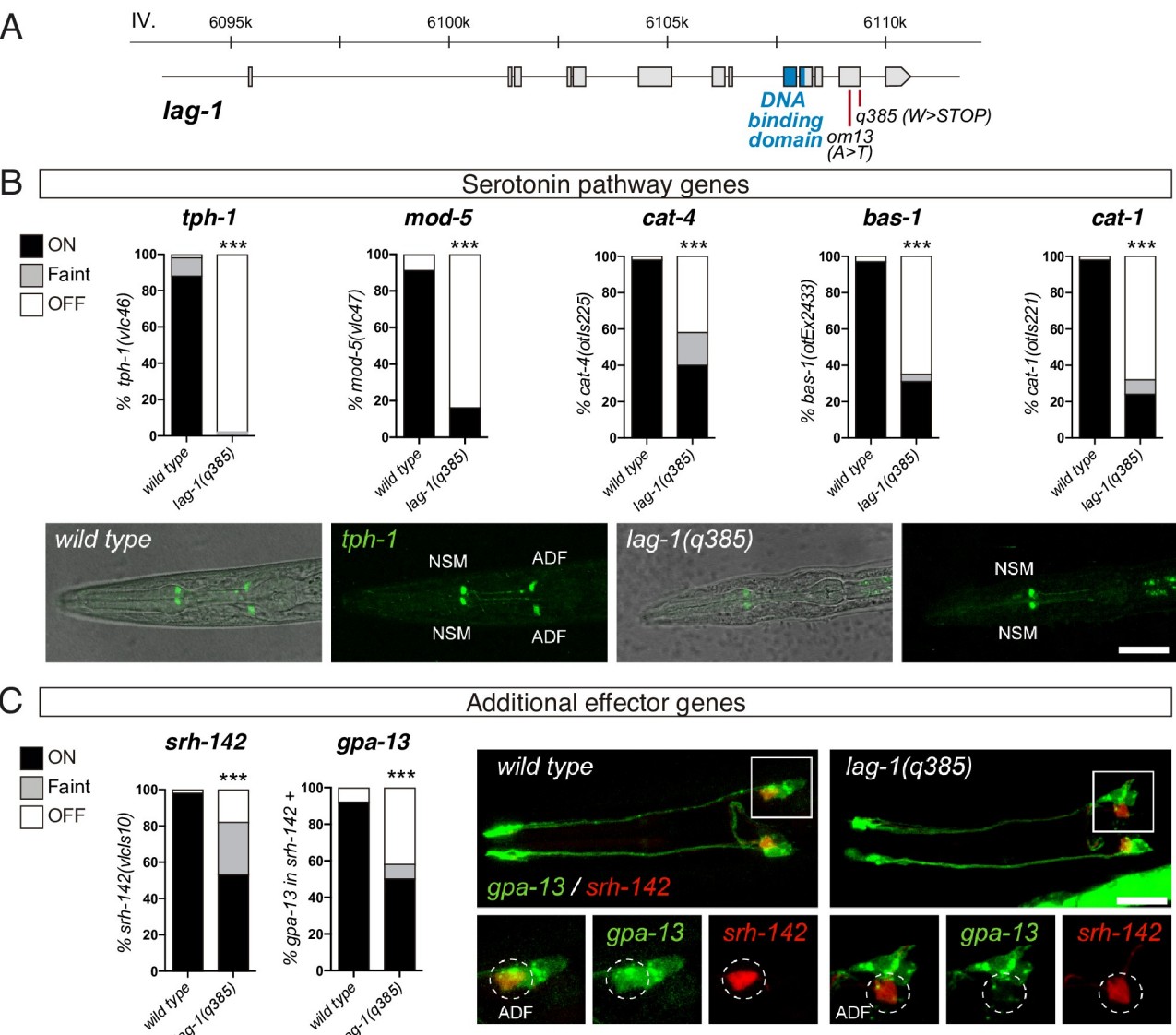

**Fig 2. LAG-1 TF controls ADF terminal differentiation.** (A) *lag-1* locus, grey boxes represent exons and blue boxes correspond to exons coding for the LAG-1 DNA binding domain. Red bars locate point mutations of the corresponding alleles. (B) Quantification of *lag-1(q385)* mutant phenotypes show significant ADF expression defects for all serotonin pathway genes. *tph-1(vlc46)* and *mod-5(vlc47)* are knock-in reporters. L1 stage images show *tph-1* expression defects in ADF. Scale bar: 15 μm. See S1 Data for numerical values. (C) *lag-1(q385)* mutant shows expression defects for additional effector genes *srh-142(vlcIs10)* and *gpa-13(pkIs589)*. For *gpa-13* expression analysis, presence of faint *srh-142* reporter expression was used to identify ADF neuron. Micrographs show faint *srh-142* and missing *gpa-13* expression in the ADF of *lag-1(q385)* mutant animals. Scale bar: 10 μm. See S1 Data for numerical values. TF, transcription factor.

[13,14]. *lag-1(q385)* individuals arrest soon after hatching in the first larval stage (L1) with numerous defects reflecting a failure of Notch-mediated cell fate specification during embryogenesis [14]. Analysis of freshly hatched *lag-1(q385)* worms revealed a dramatic loss of 5HT pathway reporter gene expression in the ADF neuron (Fig 2B). Interestingly, the 2 knock-in endogenous reporters analysed, *tph-1/tryptophan hydroxylase (vlc46)* and *mod-5/5HT transporter (vlc47)*, show the greatest expression defects compared to other scored reporters built as multicopy extrachromosomal or integrated simple arrays [*otIs221(cat-1::gfp)*, *otIs225(cat-4::gfp)*, and *vlcEx2433(bas-1::gfp)*] (Fig 2B). This observation suggests that endogenous reporters

might be more sensitive to *lag-1* loss. Expression of other effector genes not directly related to serotonin biosynthesis, such as *srh-142/serpentine receptor* and *gpa-13/Gα protein*, is also reduced in *lag-1(q385)* animals (Fig 2C), suggesting a broader role for LAG-1 in ADF neuron terminal differentiation, not limited to serotonin metabolism, as would be expected for a terminal selector. Importantly, we determined that ADF neuron is still generated in *lag-1(q385)* mutant as 82% of *lag-1(q395)* animals still show detectable *srh-142/serpentine receptor* expression albeit at lower levels (Fig 2C). We next analysed *odr-1/guanylate cyclase* effector gene expression in AWB neurons, the sister cell of ADF, and found similar *odr-1/guanylate cyclase* expression in *lag-1(q385)* mutant and wild-type animals (S2 Fig). Altogether, these results suggest that LAG-1 function is dispensable for the generation of 2 neurons in the terminal branch of the ADF lineage (ADF and AWB neurons) or for ADF neuron survival, but it is necessary for correct ADF neuron terminal differentiation, likely through direct activation of ADF expressed effector genes. Similar broad ADF expression defects are found in the thermosensitive hypomorphic allele *lag-1(om13)*, although the phenotype is observed only when worms are grown at 15 ˚C and only transiently during L1 (S2 Fig). This allele shows slight germline and somatic defects, which are stronger at 20 to 25 ˚C compared to 15 ˚C [15]. The transient phenotype we observe is likely due to the weak nature of this hypomorphic allele in which enough LAG-1 activity in the ADF neuron might be reached either at longer times or at higher temperatures. Finally, as will be discussed in more detail below, *lag-1* RNAi also displays broad serotonin pathway gene expression defects specifically in the ADF.

## LAG-1 is expressed in the postmitotic ADF neuron to induce and maintain ADF neuron fate

*Cis*-regulatory analysis and *lag-1* mutant characterisation suggested that LAG-1 acts as terminal selector for the ADF chemosensory neuron. Available single-cell RNAseq (sc-RNAseq) expression data from *C. elegans* embryos [16] reveal bilateral *lag-1* expression in the ADF lineage starting at the postmitotic ADF neuroblast, supporting a terminal role for LAG-1 (Fig 3A). Embryonic *lag-1* expression is also detected in the terminal division of 3 additional postmitotic neuroblasts RIB, AIM, and RIH. Interestingly, AIM and RIH express *mod-5/5HT transporter* and *cat-1/vesicular monoamine transporter* to reuptake 5HT from the extracellular space. In addition to ADF specification defects, *lag-1(q385)* animals show *mod-5/5HT transporter* and *cat-1/vesicular monoamine transporter* reporter expression defects in AIM and RIH (S2 Fig), suggesting LAG-1 could also play a role in the terminal differentiation of these additional neuronal types.

To assess postembryonic LAG-1 expression, we labelled the endogenous *lag-1* locus with a fluorescent reporter (*lag-1::T2A::mNeonGreen*) (Fig 3B). This construct generates a unique mRNA transcript that, upon translation, undergoes autocatalytic cleavage, giving rise to 2 independent proteins, LAG-1 and the fluorescent mNeonGreen protein, ensuring correct LAG-1 function (Fig 3B). At L1, *mNeonGreen* expression is observed in several neurons in the head including ADF neuron (Fig 3B). Strong LAG-1 expression in ADF neuron is maintained at all larval stages and throughout the life of the animal (Fig 3B). Similar strong ADF expression at L1 and young adult stages was observed for a fusion protein LAG-1::GFP using a fosmid-recombineered reporter strain (*vlcEx496*) (S3 Fig).

We next further characterised *lag-1* expression in the context of the known ADF gene regulatory network. During embryonic development, the LIM HD TF *lim-4* is expressed in the mother cell of ADF/AWB. *lim-4(yz12)* mutant animals generate 2 neurons in the terminal division of the ADF lineage, but ADF neuron fails to properly differentiate [17]. Our data indicate that LIM-4 is required for LAG-1 expression in the ADF neuron (S4 Fig).

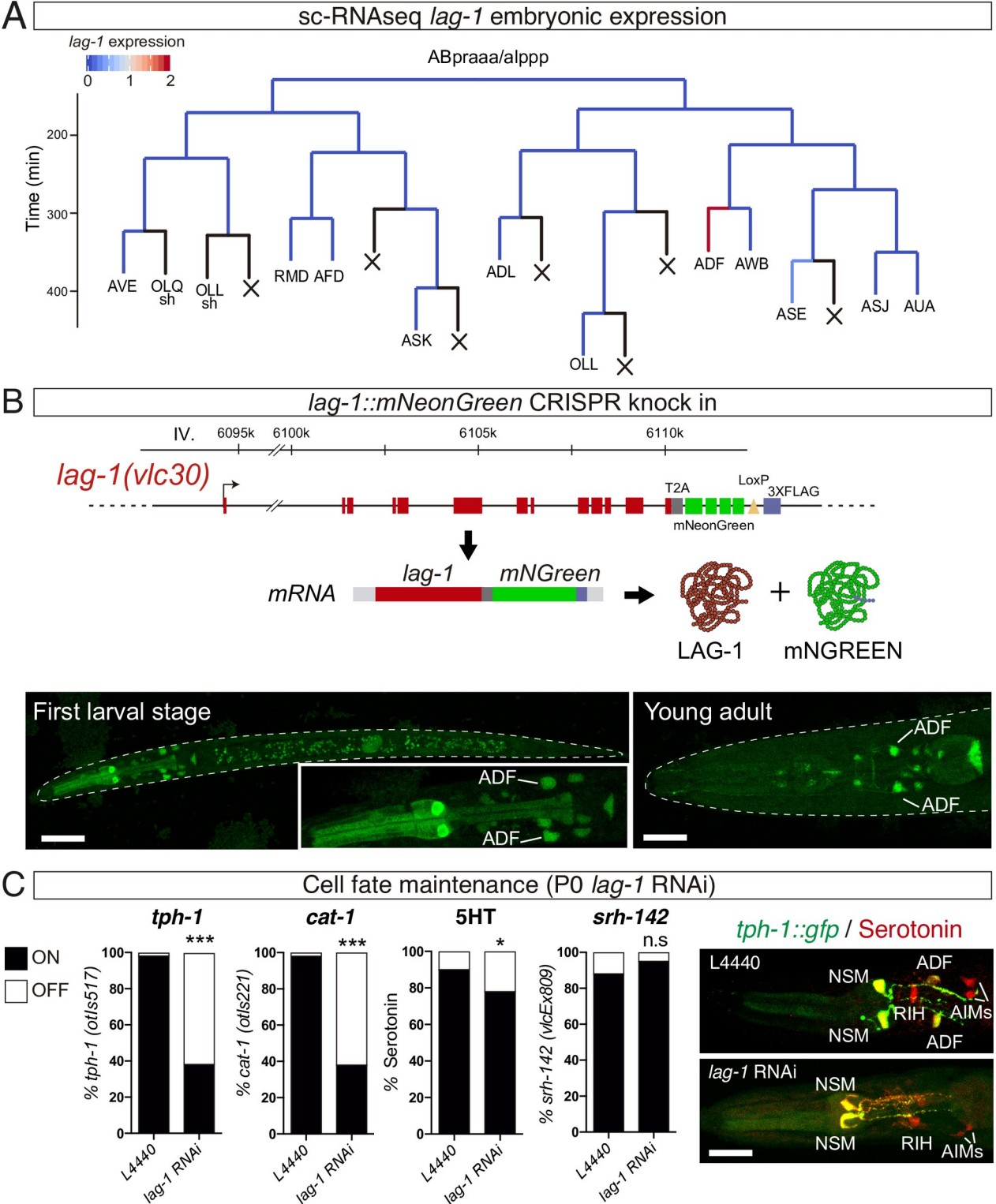

**Fig 3. LAG-1 is expressed in postmitotic ADF to induce and maintain ADF neuron fate.** (A) ADF lineage representing embryonic *lag-1* expression assessed by sc-RNAseq [16]. Black lines represent no expression data available in most cases corresponding to cells that undergo programmed cell death. (B) *lag-1* locus tagged with mNeonGreen to monitor *lag-1* expression in vivo. Red boxes represent coding exons of *lag-1*, grey box T2A sequence, green boxes mNeongreen fluorochrome, yellow triangle LoxP site, and blue box 3x flag tag. This locus gives rise to a single mRNA that undergoes autocatalytic cleavage upon translation. mNeongreen expression in the ADF is observed at L1, and expression is maintained

throughout the life of the animal. Scale bars: 20 μm and 10 μm. ADF identity was assessed by colocalization with *tph-1::dsred*. (S2 Data). C) *lag-1* P0 RNAi reveals that *lag-1* is continuously required to maintain *tph-1(otIs517)* and *cat-1(otIs221)* expression and 5HT staining in the ADF but not for *shr-142(vlcEx809)* reporter expression maintenance. Micrographs show missing *tph-1* expression and 5HT staining defects in adult ADF of *lag-1* RNAi fed worms while other serotonergic cells are unaffected. See S1 Data for raw data of all replicates. Scale bar: 12 μm. See S1 Data for numerical values. L1, first larval stage; n.s, nonsignificant; sc-RNAseq, single-cell RNAseq; 5HT, serotonin.

Terminal selectors are often required to maintain neuron fate [6]. As LAG-1 is continuously expressed in the ADF neuron, we checked whether its activity is also required for ADF cell fate maintenance. Postembryonic knockdown of *lag-1* by feeding with *lag-1* RNAi L1 *rrf-3 (pk1426)* animals, an RNAi neuron-sensitised strain, leads to sterile animals, similar to postembryonic *lin-12* and *glp-1* Notch receptors RNAi treatment. *lag-1* P0 RNAi animals show a significant loss of *tph-1/tryptophan hydroxylase*, *cat-1/vesicular monoamine transporter*, *bas-1/aromatic amino-acid decarboxylase*, and *cat-4/GTP cyclo-hydrolase* reporter expression in young adult animals, as well as 5HT staining defects (Fig 3C and S4 Fig). These defects are not due to neurodegeneration of the ADF neuron as *srh-142/serpentine receptor* reporter expression is unaffected (Fig 3C).

Next, we attempted to assess whether ADF specification defects in *lag-1* mutants can be rescued by cell autonomous LAG-1 expression under *srh-142* promoter, which is unaffected in *lag-1(om13)* hypomorphic mutants (S4 Fig). *lag-1* gene codes for 4 different splicing variants (S3 Fig), we failed to observe significant *tph-1/tryptophan hydroxylase* expression rescue with any individual isoform or with a combination of the 4 transcripts (S4 Fig). *lag-1(om13)* phenotype is only transiently observed during L1; this short time window might limit our ability to detect *lag-1* rescue; alternatively, different levels of specific isoform or earlier expression might be required for rescue. As an alternative to cell autonomous rescue, we performed cell autonomous knockdown of *lag-1* in postmitotic ADF neurons using the ADF-specific promoter *srh-142* to drive *lag-1* double-stranded RNA. Cell type–specific *lag-1* RNAi depletes endogenous LAG-1 specifically from the ADF neuron (Fig 4A), and this depletion is sufficient to induce expression defects of endogenous *tph-1/tryptophan hydroxylase* in the ADF (Fig 4A). These data support a cell autonomous role for LAG-1 in ADF terminal differentiation.

Finally, we assessed if LAG-1 is not only necessary but also sufficient to induce ADF fate. We used a heat shock–inducible promoter to broadly express LAG-1 in embryos at the stage of neurogenesis. LAG-1 isoform D, but not LAG-1 isoform A, is sufficient to drive ectopic *tph-1* reporter expression (Fig 4B), suggesting LAG-1 is sufficient in some contexts to induce ADF gene expression and unravelling isoform specific functions for LAG-1. Both isoforms share the CSL DNA binding domain and the Notch interacting domain, located at the carboxyl terminal region of the protein, but differ in the amino-terminal region, which has unknown functions.

Altogether, our data strongly indicate that LAG-1 acts as terminal selector to directly induce and maintain expression of ADF effector genes.

## Physiological processes dependent on ADF neuron require postembryonic LAG-1 function

5HT release from ADF chemosensory neuron mediates a variety of processes and behaviours. Thus, we next explored if postembryonic *lag-1* function is required for correct display of ADF-regulated behaviours. To circumvent larval lethality of *lag-1(q385)* null mutants, we took advantage of postembryonic knockdown of *lag-1* by RNAi in the *rrf-3(pk1426)* sensitised strain.

ADF has been implicated in pharyngeal pumping in well-fed animals [18]; however, this role is unclear as other reports implicate 5HT release from NSM and not ADF as the main

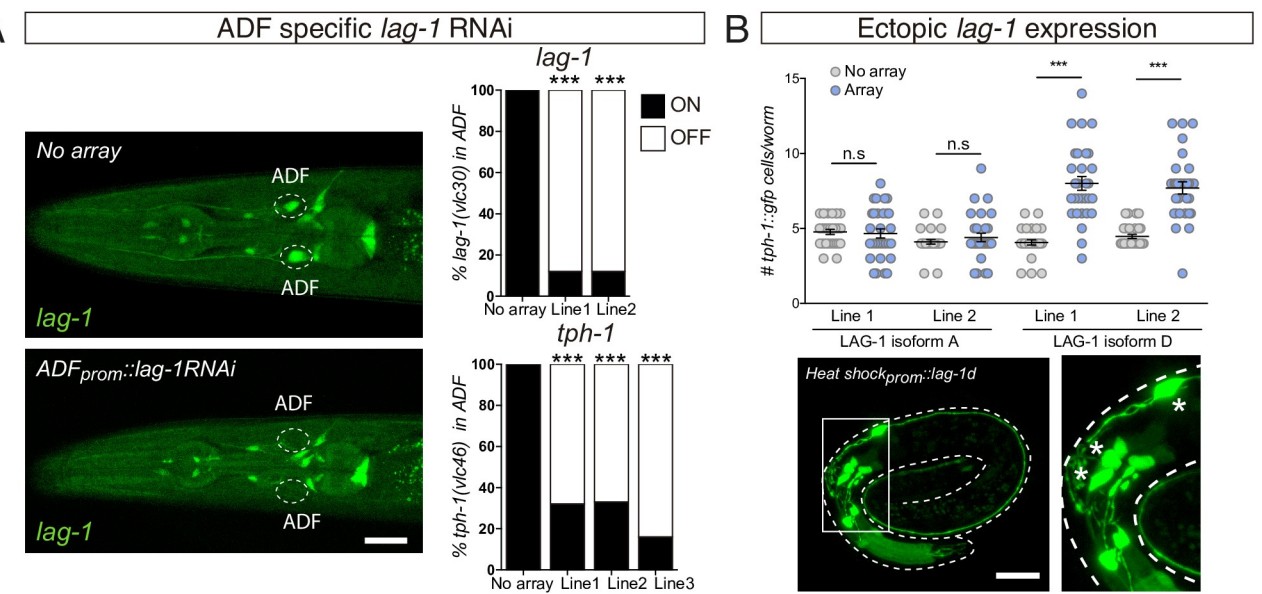

**Fig 4. LAG-1 acts cell autonomously in ADF, and it is sufficient to induce ADF effector gene expression.** (A) ADF-specific *lag-1* knockdown expressing sense and antisense *lag-1* under the *srh-142* promoter reduces endogenous *lag-1(vlc30)* and *tph-1(vlc46)* expression only in ADF, supporting the cell autonomous effect of LAG-1. Young adult images show the specificity of the *lag-1* RNAi. Scale bar: 10 μm. See S1 Data for numerical values. (B) Embryonic ectopic expression of 2 different *lag-1* isoforms using a heat shock–inducible reporter. LAG-1D isoform induces ectopic *tph-1(zdIs13)* reporter expression, while LAG-1A isoform has no effect. Similar effects are observed for 2 independent transgenic lines. n=30. Mean and SEM are represented. Representative micrograph of a *lag-1d* heat shocked embryo shows *tph-1(zdIs13)* reporter expression in 7 cells in the head. Asterisks mark ectopic GFP expressing cells. Scale bar: 10 μm. See S1 Data for numerical values. n.s, nonsignificant.

regulator of pharyngeal pumping [19]. We find that *lag-1* RNAi-fed *rrf-3(pk1426)* worms show no difference in pumping rates compared to L4440 RNAi-fed animals (Fig 5A), suggesting LAG-1 function is not required, at least under these conditions, to modulate pharyngeal pumping.

ADF, together with ASI and ASG chemosensory neurons, is required to prevent dauer entry [20], and, accordingly, *tph-1/tryptophan hydroxylase* mutant animals show enhanced dauer entry [21,22]. We find a significant increase in the percentage of dauer animals upon pheromone treatment in P0 *lag-1* RNAi-fed worms compared to controls (Fig 5B), suggesting LAG-1 is required to prevent dauer entry. Of note, the *lag-1(om13)* hypomorphic allele, which shows no obvious ADF specification defects after L1, has been previously used to show a role for *lag-1* in dauer maintenance and dauer exit, a process that is mediated through expression of the Notch ligand *lag-2* in the IL2 neurons, but a role for ADF neuron in this processes has not been reported [23].

5HT release from ADF neuron also promotes fat mobilisation [24]; accordingly, we find that *lag-1* RNAi induces a significant increase in lipid storage (Fig 5C), a phenotype also observed in *tph-1/tryptophan hydroxylase* mutants [24]. In addition, ADF function is also involved in foraging: after a food deprivation period, when worms are approaching food source, they reduce their speed [25]. This behaviour is mediated by 5HT release from NSM and ADF neurons [26]. We find that postembryonic knockdown of *lag-1* by RNAi does not affect basal locomotion, but *lag-1* RNAi-fed animals fail to decelerate upon food detection prior to contact (Fig 5D and 5E). As neither *lag-1* RNAi nor *lag-1(q385)* mutants show NSM serotonergic specification defects, our results suggest that postembryonic LAG-1 function in ADF is required for correct foraging behaviour.

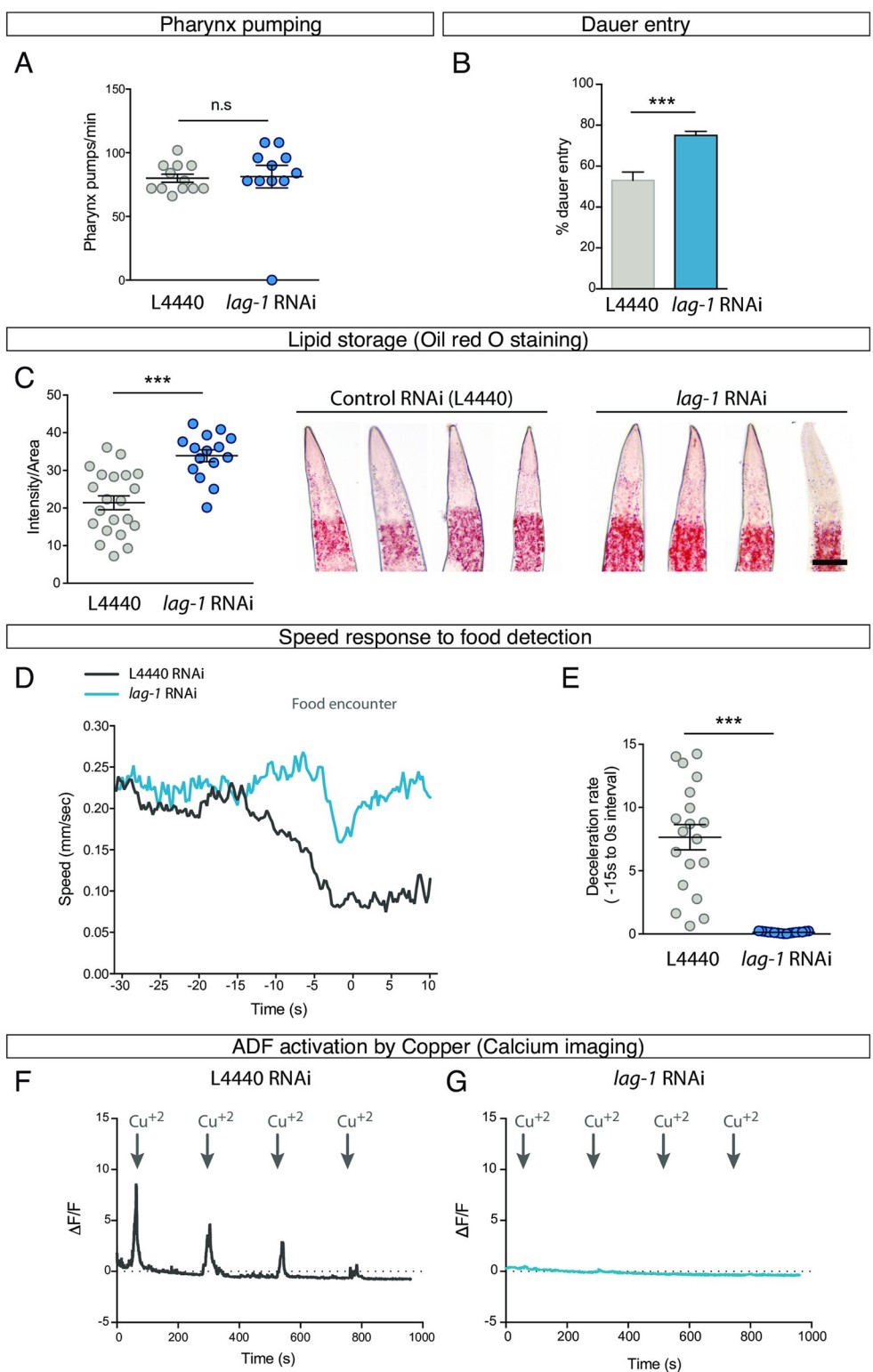

**Fig 5. ADF-mediated processes require postembryonic LAG-1 function.** (A) *lag-1* P0 RNAi-fed worms show no defects in pharyngeal pumping rate. $n \geq 10$. Mean and SEM are represented. See S1 Data for raw data of all replicates. See S1 Data for numerical values. (B) *lag-1* P0 RNAi-fed worms show a significant increase in dauer induction by pheromones compared to control RNAi. $n = 148$ worms in controls and $n = 474$ in *lag-1* RNAi. Percentage and SEP are represented. See S1 Data for raw data of all replicates. See S1 Data for numerical values. (C) *lag-1* P0 RNAi-fed

worms contain significantly higher levels of lipids stained by oil red. See S1 Data for raw data of all replicates; $n \geq 15$ per replicate. Scale bar: 13 μm Mean and SEM are represented. See S1 Data for numerical values. (D) Quantification of worm's speed after a short starvation period when approaching a patch of food in *lag-1* P0 RNAi and control animals. Curves represent the mean value for each condition. $n = 19$ worms in controls; $n = 17$ worms in *lag-1* RNAi. See S1 Data for numerical values. (E) Mean deceleration rate during 15 seconds prior to food encounter shows significant differences between *lag-1* RNAi and control animals. $n = 19$ worms in L4440; $n = 17$ worms in *lag-1* RNAi. Two-tailed *t* test. Mean and SEM are represented. See S1 Data for numerical values. (F) Calcium imaging quantification in control worms shows that ADF is activated by $Cu^{+2}$ and progressively adapts to the stimulus. The curve represents the mean value of analysed worms. $n \geq 15$ animals were recorded. See S1 Data for numerical values. (G) ADF is not activated by $Cu^{+2}$ in *lag-1* P0 RNAi-fed animals. The curve represents the mean value of analysed worms. $n \geq 15$ animals were recorded. See S1 Data for numerical values. n.s, nonsignificant.

Finally, we directly tested ADF activation in *lag-1* RNAi-fed worms by calcium imaging. As expected, ADF L4440-fed control RNAi animals respond to short exposures to $Cu^{+2}$ solution [27] (Fig 5F). In sharp contrast, the same stimulus fails to activate the ADF neuron in worms fed with *lag-1* RNAi (Fig 5G). Thus, LAG-1 function is required postembryonically to either allow for copper detection or to elicit correct activation in response to this stimulus.

Altogether, our physiological and behavioural assays show that postembryonic LAG-1 function is required not only for 5HT pathway gene expression but also for correct ADF activation and proper performance of ADF-mediated processes.

## LAG-1 activating functions in ADF neuron are Notch independent

LAG-1 is the signal-regulated transcriptional mediator of the phylogenetically conserved Notch intercellular signalling pathway [13]. Accordingly, we next aimed to explore if depletion of other components of the Notch pathway produce similar ADF specification defects. *C. elegans* genome codes for 2 Notch receptors GLP-1 and LIN-12, which, upon activation, are proteolytically cleaved, allowing the intracellular domain to translocate to the nucleus (Fig 6A). In the nucleus, the Notch intracellular domain (NICD) forms a complex with LAG-1 and the coactivator SEL-8 to shift the basal repressor activity of LAG-1 towards transcriptional activation of target genes [28]. Previously reported phenotypes for *lag-1* are mimicked by *glp-1*, *lin-12*, and *sel-8* mutants [14,29]. Accordingly, *rrf-3(pk1426)* L1 larvae fed with *glp-1*, *lin-12*, or *sel-8* RNAi develop into young adults that are sterile or produce dead embryos, similar to *lag-1* RNAi-fed animals. However, in contrast to *lag-1* RNAi, these animals do not show defects in *tph-1/tryptophan hydroxylase* or *cat-1/vesicular monoamine transporter* reporter expression (Fig 6B and 6C). We next analysed ADF fate in *lin-12(n941) glp-1(q46)* double mutants and the *sel-8(ok387)* single mutant, which should abrogate all zygotic Notch pathway activity. Similar to *lag-1(q385)* null homozygotes, these animals die at L1; however, contrary to *lag-1* mutants, L1 larvae show normal *tph-1/tryptophan hydroxylase* reporter expression in the ADF neuron (Fig 6B and 6C).

*glp-1* is required maternally for early embryonic development [30]. To bypass this requirement and test the possibility that maternally supplied *glp-1* in *lin-12(n941) glp-1(q46)* double mutants could be contributing to ADF terminal differentiation, we used a temperature-sensitive allele of *glp-1*. *glp-1(e2144)* gravid mothers, which contain embryos that have passed the stage where maternal *glp-1* is required, were placed at restrictive temperature and allowed to lay eggs. Eggs are grown at restrictive temperature since approximately 30-cell stage, which corresponds to 4 divisions prior ADF birth. As expected from *glp-1* depletion, these animals are sterile; however, they do not display *tph-1/tryptophan hydroxylase* reporter expression defects in the ADF neuron at L1 or as young adult (Fig 6B). Finally, we were unable to detect ADF expression of *lin-12* or *glp-1* at L1 or young adult using both endogenous knock-in and fosmid-based fluorescent reporters (S5 Fig). Similarly, available embryonic or L2 sc-RNAseq

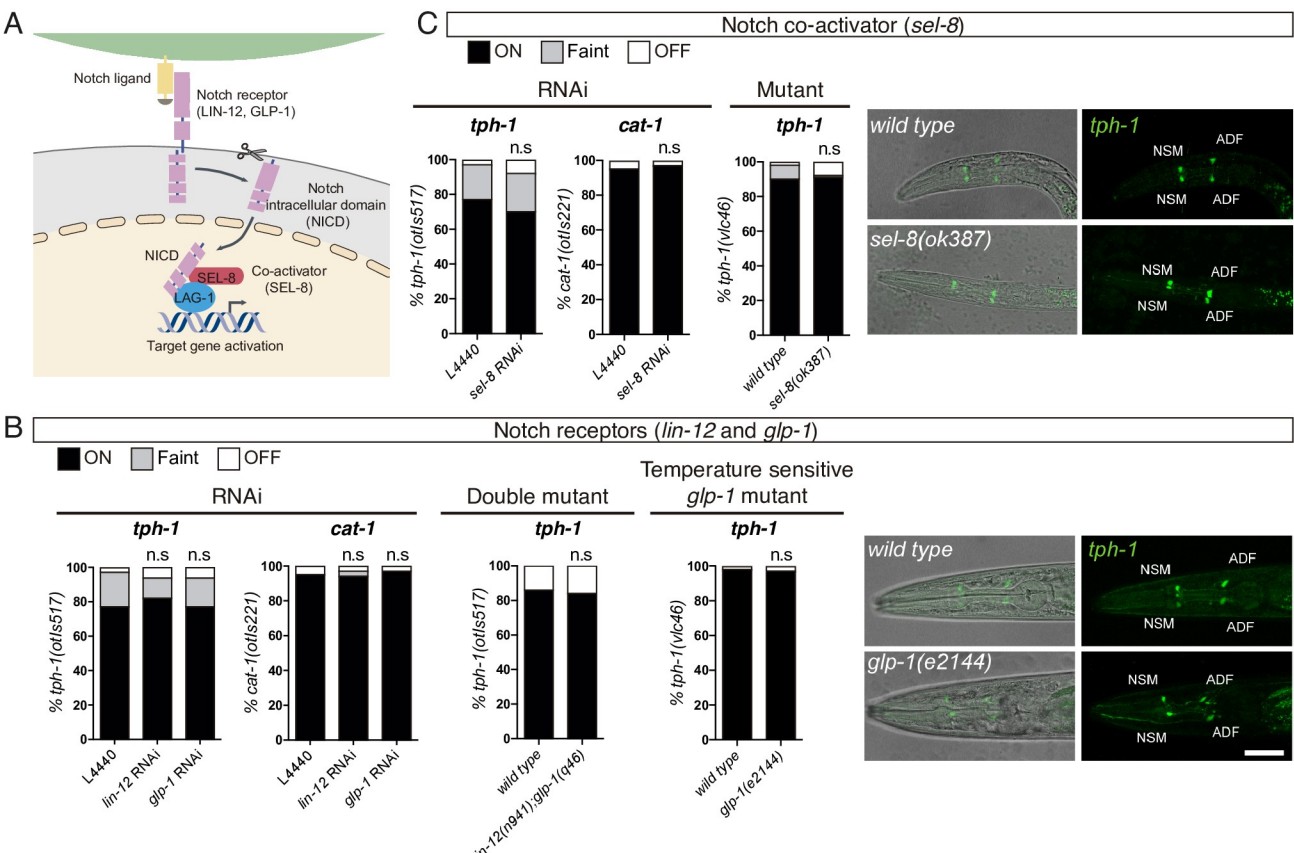

**Fig 6. *glp-1* and *lin-12* Notch receptors and *sel-8* coactivator are not required for ADF specification.** (A) Schematic representation of Notch signalling pathway in *C. elegans*. (B) P0 RNAi and double mutant analysis for *glp-1* and *lin-12* Notch receptors and temperature-sensitive allele analysis for *glp-1* show no significant difference in *tph-1(otIs517)* and *cat-1(otIs221)* reporter expression in the ADF neuron. L1 stage confocal images showing representative examples of the temperature shift assay. Scale bar: 15 μm. See S1 Data for numerical values of all replicates and temperature-sensitive allele scoring at YA stage. (C) P0 RNAi and mutant analysis of *sel-8* coactivator shows no difference in *tph-1(otIs517)* and *cat-1(otIs221)* reporter expression in the ADF neuron. L1 stage confocal images showing representative examples. See S1 Data for numerical values. L1, first larval stage; NICD, Notch intracellular domain; n.s, nonsignificant; YA, young adult.

data do not show *sel-8*, *glp-1*, or *lin-12* expression in the ADF or the ADF mother cell [16,31] (S5 Fig). Altogether, these data strongly suggest that LAG-1 functions as activator of ADF effector genes are independent of the canonical Notch pathway.

Notch-independent activating functions for CSL TFs have been previously reported. In mammals, Rbpj/CSL specifies mouse pancreatic cells and cerebellar and spinal cord GABAergic neurons through its interaction with the bHLH TF Ptf1a [32,33]. These data prompted us to explore the possibility that LAG-1 acts together with HLH-13, *C. elegans* ortholog of Ptf1a, in the specification of the ADF neuron. However, *hlh-13(tm2279)* mutants show normal 5HT pathway gene expression in ADF neurons (S6 Fig). In mammals, Rbpj, Ptf1a, and an E protein form a trimeric complex that binds a DNA motif composed by an Ebox and a CSL site [33,34]. Thus, we reasoned that, if LAG-1 interacts similarly with other unknown TFs, a composite site "CSL+ unknown DNA motif" might be present in the regulatory regions of ADF effector genes. We thus performed sequence alignments with all experimentally determined functional CSL binding sites +/− 30 bp of flanking sequences in *C. elegans* and other *Caenorhabditis* species. We retrieved 66 conserved CSL binding sites from 19 different species (S6 Fig). Alignment of all these sequences does not reveal any additional motif flanking the CSL site (S6 Fig),

which might be due to flexibility in the position and/or orientation of the additional motif. Thus we performed de novo motif analysis using this set of 66 sequences, which retrieved, in addition to CSL binding site, 2 motifs that show some similarity to bHLH TF binding motifs (S6 Fig). *C. elegans* genome codes for 41 bHLH TFs in addition to *hlh-13*, it is possible that other bHLH TF could work together with LAG-1 to induce ADF fate. Further studies should be aimed to identify such factor.

## LAG-1 regulates ADF *tph-1/tryptophan hydroxylase* expression plasticity upon pathogen infection and cilia defects

ADF is a plastic sensory neuron that, in response to several insults, up-regulates expression of *tph-1/tryptophan hydroxylase*, the rate-limiting enzyme for 5HT biosynthesis, increasing 5HT signalling. Thus, we tested the role of LAG-1 in *tph-1/tryptophan hydroxylase* expression plasticity in the ADF neuron.

In agreement with previous reports based on *tph-1/tryptophan hydroxylase* transgenic reporter strains [22,35,36], we found that dauer entry, heat stress, and *unc-43*/CAMKII gain-of-function mutation (a mediator of neuronal activity and synaptic plasticity) induce expression of the endogenous *tph-1/tryptophan hydroxylase* locus (*vlc46*) (Fig 7A, 7C, and 7D). A similar increase in expression is observed in dauer animals from *lag-1* RNAi-fed worms, in heat shocked *lag-1(om13)* or *unc-43(n498*gof); *lag-1(om13)* double mutants animals (Fig 7B–7D), suggesting *tph-1/tryptophan hydroxylase* expression plasticity under these conditions is independent of LAG-1 function.

*tph-1/tryptophan hydroxylase* transcriptional reporter expression is also induced in animals with cilia morphology defects [22]. In agreement with these data, we find that endogenous *tph-1/tryptophan hydroxylase* expression (*vlc46*) is induced in mutants that have cilia defects, *dyf-1(yz66)* and *che-11(e1810)* (Fig 7E and 7F). Importantly, this induction is not observed when LAG-1 function is compromised with the *lag-1(om13)* allele (Fig 7E and 7F), indicating that, in contrast to dauer state, heat stress, or CAMKII overactivation, LAG-1 mediates *tph-1* expression plasticity induced by cilia morphology defects.

Finally, we analysed *tph-1/tryptophan hydroxylase* induction upon pathogenic infection. Pathogenic bacteria *Pseudomonas aeruginosa* induces *tph-1/tryptophan hydroxylase* reporter expression in the ADF neuron; this increase in transcription is mediated by Toll and Interleukin 1 Receptor (TIR-1) [37–39]. In agreement with previous reports with transcriptional reporters [37–39], we find that a gain-of-function allele *tir-1(yz68)* shows increased expression of the endogenous *tph-1/tryptophan hydroxylase* locus (*vlc46*) (Fig 7G). We failed to generate viable *tir-1(yz68); lag-1(om13)* double mutants; therefore, we performed *lag-1* RNAi in *tir-1 (yz68); rrf-3(pk1416)* animals. *lag-1* RNAi abolishes *tph-1/tryptophan hydroxylase* reporter induction (Fig 7H), suggesting that, similar to cilia morphology defects, LAG-1 is necessary to mediate ADF plastic response to pathogenic bacteria. Similar to its role as terminal selector, LAG-1 function on *tph-1/tryptophan hydroxylase* expression plasticity is also Notch independent as neither *glp-1* nor *lin-12* are necessary for *tph-1/tryptophan hydroxylase* induction (S7 Fig).

DAF-19, the only RFX TF in *C. elegans*, is required for developmental *tph-1/tryptophan hydroxylase* expression as well as to mediate *tph-1/tryptophan hydroxylase* transcriptional increase upon different insults including bacterial infection or cilia morphology defects. However, DAF-19 does not regulate expression of other ADF 5HT effector genes [38]. DAF-19 has been suggested to act indirectly on *tph-1* expression through yet unidentified factors [38]. We find that ADF LAG-1 expression in *daf-19(m86)* mutants shows no difference compared to controls (S8 Fig), indicating *lag-1* transcription is not downstream of DAF-19.

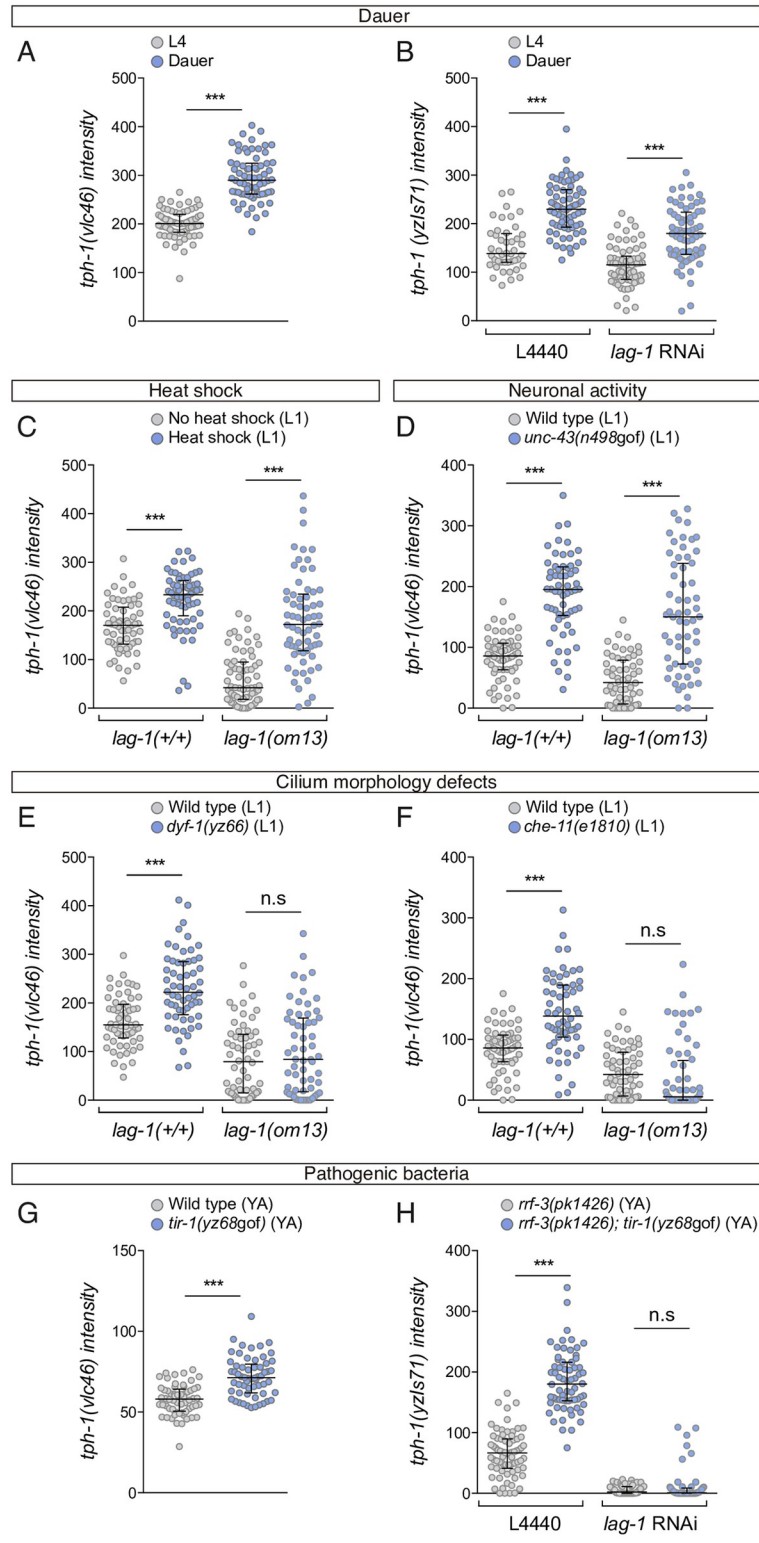

**Fig 7. LAG-1 mediates *tph-1* expression induction by specific stimuli.** (A) ADF mNeonGreen intensity quantification of *tph-1(vlc46)* endogenous reporter in dauer stage shows a significant increase compared to L4. *n* > 60 neurons. Median and IQR are represented. See S1 Data for numerical values. (B) Dauer animals show increased ADF *yzIs71(tph-1::gfp)* reporter expression both in *lag-1* RNAi and control animals, suggesting that LAG-1 does not mediate *tph-1* induction in dauer. Median and IQR are represented. See S1 Data for numerical values. (C) L1 ADF

mNeonGreen intensity quantification of *tph-1(vlc46)* upon heat shock. Heat shock induces *tph-1* expression both in wild-type and *lag-1(om13)* mutant worms, suggesting that *lag-1* is not required in the heat response. Median and IQR are represented. See S1 Data for numerical values. (D) L1 ADF mNeonGreen intensity quantification of *tph-1(vlc46)* endogenous reporter in *unc-43(n498)*/CAMKII gain of function mutation. *unc-43(n498gof)* induces *tph-1* expression both in wild-type and *lag-1(om-13)* mutants, suggesting that *lag-1* is not required to mediate response to neuronal activation. Median and IQR are represented. See S1 Data for numerical values. (E) L1 ADF mNeonGreen intensity quantification of *tph-1(vlc46)* endogenous reporter in *dyf-1(yz66)* mutants shows increased levels of *tph-1* expression that are not observed in *dyf-1(yz66); lag-1(om13)* double mutants, suggesting that *lag-1* mediates *tph-1* expression response to cilia damage. Median and IQR are represented. See S1 Data for numerical values. (F) Similar *lag-1* dependency of *tph-1(vlc46)* endogenous reporter induction is observed in an additional intraflagellar transport protein mutant, *che-11(e1810)*. Median and IQR are represented. See S1 Data for numerical values. (G) Young adult ADF mNeonGreen intensity quantification of *tph-1(vlc46)* endogenous reporter in *tir-1(yz68*gof*)* gain-of-function mutation, which mimics response to pathogenic bacteria. Median and IQR are represented. See S1 Data for numerical values. (H) *tir-1(yz68*gof*)* induction of *yzIs71(tph-1::gfp)* reporter is not observed in animals treated with *lag-1* RNAi, suggesting that *lag-1* mediates *tph-1* expression response to pathogenic bacteria. Median and IQR are represented. See S1 Data for numerical values. IQR, interquartile range; n.s., nonsignificant; YA, young adult.

## Different *cis*-regulatory elements mediate *tph-1/tryptophan hydroxylase* expression plasticity to external stimuli

The mechanistic details of how plastic gene expression responses are integrated at the *cis*-regulatory level are still very scarce. Thus, we next analysed if *tph-1/tryptophan hydroxylase* expression modulation is mediated through the developmental minimal CRM (*tph-1p17*) that contains a functional LAG-1/CSL binding site (Fig 8A). Interestingly, as explained in detail below, our results reveal that a complex variety of *cis*-regulatory mechanisms are used to integrate responses to different stimuli.

*unc-43*/CAMKII gain-of-function mutation, which induces endogenous *tph-1/tryptophan hydroxylase* independently of LAG-1 (Fig 7D), does not induce activation through *tph-1prom17* (Fig 8B), suggesting that *cis*-regulatory elements outside this developmental CRM mediate the plastic gene expression response to neuronal activity. Unexpectedly, *tir-1(yz68)* gain-of-function allele, which requires LAG-1 for endogenous *tph-1/tryptophan hydroxylase* induction (Fig 7H), does not activate *tph-1prom17* (Fig 8C), suggesting that the response to pathogenic bacteria is also integrated outside the developmental minimal CRM. One additional CSL binding site is present in the intergenic region of *tph-1/tryptophan hydroxylase*, which is located immediately outside of *tph-1prom17* CRM (S8 Fig). In contrast to the absolute requirement of the CSL binding site in *tph-1prom17*, mutation of this site in a context of a longer *tph-1* CRM (*tph-1prom2*) that contains both CSL sites produce only a partial loss of GFP expression (S8 Fig). Only when both CSL sites are mutated GFP expression is completely abolished, unravelling the functionality of this additional CSL site (S8 Fig). In contrast to lack of *tph-1prom17* activation, *tph-1prom2* expression is induced in *tir-1(yz68)* gain-of-function animals. *tir-1(yz68)* induction of *tph-1prom2* is completely abolished when both CSL sites are mutated (S8 Fig). Thus, LAG-1 mediates *tph-1* gene induction in the presence of pathogenic bacteria through both CSL binding sites.

In contrast, dauer state, heat stress, and cilia morphology defects enhance *tph-1/tryptophan hydroxylase* expression through the activation of *tph-1prom17* minimal CRM (Fig 8D–8F). As we had previously determined that heat stress and dauer stage induction of endogenous *tph-1/tryptophan hydroxylase* act independently of LAG-1 (Fig 7A–7C), we hypothesised that the response to these stimuli might be integrated through additional non-CSL binding sites. In agreement with this hypothesis, we find that neither heat stress nor dauer state increase *tph-1prom46* transcription, the synthetic reporter composed by 3 CSL binding motifs that is specifically expressed in the ADF, suggesting that these stimuli act on additional TF binding sites present in the *tph-1prom17* minimal CRM (Fig 8D and 8E). Interestingly, CSL mutation in the

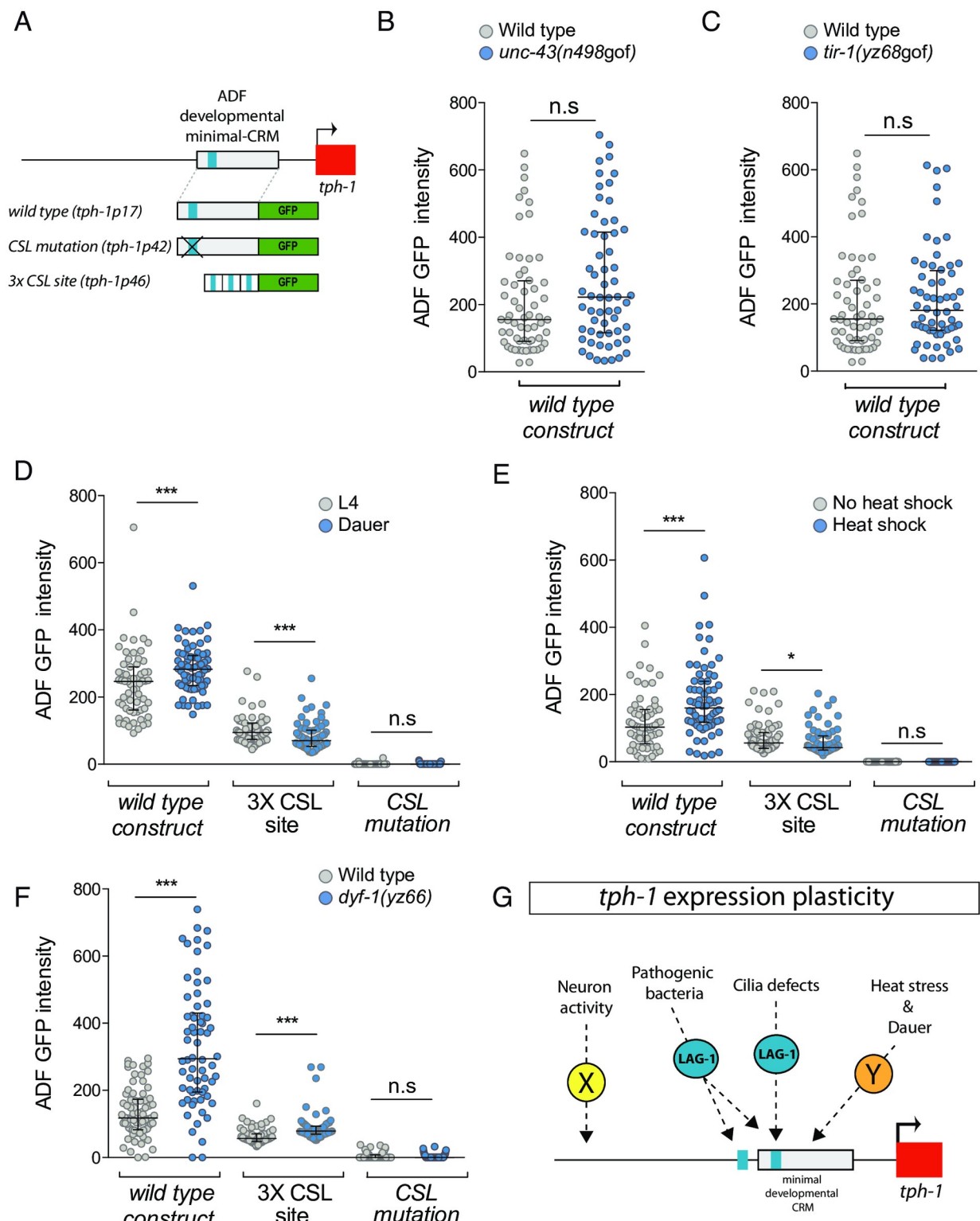

**Fig 8. Different *tph-1/tryptophan hydroxylase cis*-regulatory regions mediate plastic responses to specific stimuli.** (A) Schematic representation of analysed reporter constructs. (B) *unc-43(n498*gof*)* mutation does not activate *tph-1prom17*, ADF minimal CRM, in young adult animals. Median and IQR are represented. See S1 Data for numerical values. (C) *tir-1(yz68*gof*)* mutation does not activate *tph-1prom17* in young adult animals. Median and IQR are represented. See S1 Data for numerical values. (D) Dauer stage induces transcription of *tph-1prom17* but not *tph-1prom46* 3XCSL synthetic enhancer. Mutation of the CSL binding site (*tph-1prom42*) completely abolishes GFP expression under normal

conditions and precludes its induction in dauer. Median and IQR are represented. See S1 Data for numerical values. (E) Heat shock induces transcription of the ADF minimal CRM *tph-1prom17* in L1 worms. *tph-1prom46* 3XCSL synthetic enhancer is not activated upon heat shock stress. Mutation of the CSL binding site completely abolishes GFP expression under normal conditions and precludes induction upon heat shock. Median and IQR are represented. See S1 Data for numerical values. (F) *dyf-1(yz66)* mutation induces transcription of the ADF minimal CRM *tph-1prom17* and *tph-1prom46* 3XCSL synthetic enhancer in young adult animals. Mutation of the CSL binding site completely abolishes GFP expression under normal conditions and precludes induction in *dyf-1(yz66)* mutants. Median and IQR are represented. See S1 Data for numerical values. (G) Summary of signalling mechanisms underlying *tph-1/tryptophan hydroxylase* expression plasticity. CRM, *cis*-regulatory module; CSL, CBF-1, Su (H), LAG-1; IQR, interquartile range; n.s, nonsignificant.

minimal CRM (*tph-1prom42*) not only abolishes transcription in basal conditions but also prevents enhancement of expression upon heat stress or in dauers (Fig 8D and 8E), indicating that a basal level of activity in the CRM might be required to allow for the integration of plastic responses to external stimuli later in the life of the animal, even if they are mediated independently of LAG-1.

Finally, we find that *dyf-1(yz66)* cilia morphology mutant, which induces endogenous *tph-1/tryptophan hydroxylase* transcription in an LAG-1–dependent manner (Fig 7G), not only activates *tph-1prom17* minimal CRM but also enhances transcription of the 3XCSL synthetic construct (Fig 8F), suggesting that information from this stimulus is integrated through the developmentally activated CSL binding site. *dyf-1(yz66)* mutants fail to induce transcription of *tph-1* minimal CRM baring a mutation of the CSL binding site (*tph-1prom42*) (Fig 8F) further supporting a direct role for LAG-1 mediating these stimuli.

In summary, we find that, in addition to its developmental role as ADF terminal selector, LAG-1 is actively involved in the regulation of ADF gene expression plasticity. Our results unveil a complex scenario with a combination of LAG-1 dependent and independent gene expression responses. LAG-1–independent integration of some external stimuli (heat stress and dauer) is mediated through the developmental minimal CRM while other stimuli (neuronal activity) act outside this minimal regulatory region (Fig 8G). Seemingly, LAG-1–mediated responses are also specific to stimulus. While cilia defects act through the CSL site present in the developmental enhancer, information regarding pathogenic bacteria is integrated through 2 CSL binding sites, one of them located outside the minimal developmental enhancer (Fig 8G).

## Discussion

In this work, we describe the role of the CSL TF family in neuronal terminal differentiation in *C. elegans*. We found that LAG-1, the signal-regulated TF mediator of Notch signalling, induces and maintains the terminal differentiation program of the ADF serotoninergic neuron. Our results show that the enduring activating action that LAG-1 exerts in ADF gene expression and function is achieved independent of Notch activity. LAG-1 role as an activator independent of Notch signalling contrasts previous reports on LAG-1 functions.

### Analysis of coregulated *cis*-regulatory elements allows the identification of novel terminal selectors

Terminal selectors coregulate transcription of multiple effector genes; therefore, regulatory regions of their target genes share the corresponding regulatory signature. We studied the CRMs of the 5HT pathway genes active in the ADF and identified the common presence of functional CSL binding sites, leading us to the unexpected identification of LAG-1 as a terminal selector for the ADF neuronal fate.

Previously identified terminal selectors have been mostly isolated from forward genetic screens. However, the null phenotype of *lag-1* is larval or embryonic lethal [14], and we found

that *lag-1(om13)*, a partially lethal hypomorphic allele, does not display ADF phenotypes in adult animals. These facts likely explain why *lag-1* alleles have not been previously isolated from forward genetic screen for ADF specification defects [22,35,38,40,41]. Additional terminal selectors with lethal phenotypes have been also identified using similar *cis*-regulatory approaches [11], which underscore the complementarity of forward genetics and *cis*-regulatory analysis. Our results not only extend the number of *C. elegans* neuron types with an assigned terminal selector but more importantly expand the number of different TF families that act as terminal selectors.

## LAG-1 terminal selector action is independent of Notch signalling

*C. elegans* LAG-1 has been shown to act as a basal repressor in the absence of Notch signalling [42,43]. Here, we describe Notch-independent functions for LAG-1 as an activator. Most probably, these Notch-independent LAG-1 activating functions in the ADF neuron are mediated in concert with unidentified tissue-specific cofactors, as discussed further below.

In addition, we find that LAG-1 Notch-independent activating functions are continuously required to maintain correct gene expression in the ADF. Postembryonic removal of LAG-1 does not only affect reporter expression and 5HT staining but also leads to broad functional defects in ADF-mediated physiological processes, exposing the extensive actions of LAG-1 in ADF gene expression and function. Finally, in some contexts, LAG-1 is also sufficient to drive ADF effector gene expression. Altogether, LAG-1 shows all known attributes for a terminal selector: cell fate induction, direct action on effector genes, and cell fate maintenance [8]. In addition to activating roles for terminal selectors, in some contexts, these TFs have been shown to also act as repressors of alternative cell fates [44–46]; although our work has not deeply explored this possibility for LAG-1, *lag-1* mutants do not up-regulate expression of *odr-1* reporter in the ADF neuron (*odr-1* is an effector gene for AWB, the sister cell of ADF), suggesting that *lag-1* acts mainly activating gene expression rather than repressing alternative fates.

## LAG-1 modulates ADF plastic gene expression responses to specific external stimuli

In some cell types, the implementation of robust and stable cell identities during development must be combined with additional mechanisms that ensure adaptive responses to specific external stimuli. How robustness and plasticity of gene expression are both integrated in the cell is currently not well understood. Terminal selectors, working in concert with other transcriptional activators and repressors, are involved in ON/OFF gene switches that take place upon dauer entry [47] or in the acquisition of sexually dimorphic gene expression [48]. In addition to these ON/OFF switches, some gene plasticity responses require subtle modulation of gene expression. Here, we have uncovered the role of a terminal selector in tuning gene expression in response to external stimuli.

Previous work in *C. elegans* identified a great variety of stimuli that increase *tph-1/tryptophan hydroxylase* transcriptional levels. These stimuli work through different parallel pathways; however, direct transcriptional mediators of *tph-1/tryptophan hydroxylase* expression plasticity had not been described [22,35,36,38]. Here, we find that transcriptional response to specific stimuli (cilia morphology defects and pathogenic bacteria), but not to others (dauer, heat stress, and neuronal activity), requires LAG-1 function. This specificity is important because it reveals a complex scenario in which terminal selector function is not always necessary to modulate gene expression and additional TFs could be specifically devoted to mediate gene expression plasticity.

### Phylogenetically conserved Notch-independent role for CSL TFs in neuronal terminal differentiation

Notch-independent activating functions for CSL TFs have been previously reported for Su(H) in Drosophila mechanosensory socket cell fate maintenance [49] and for *Rbpj* in the specification of mouse pancreatic cells, and cerebellar and spinal cord GABAergic neurons [32,33]. The mechanism by which Su(H) acts as an activator in Drosophila is unknown, but in mammals, Rbpj interacts through its NICD interacting domain with the bHLH TF Ptf1a. We find that *hlh-13*, *C. elegans* ortholog of Ptf1a, is not required for correct *tph-1/tryptophan hydroxylase* reporter expression in the ADF. Nevertheless, the formal logic that has been described in mammalian cells may still operate in *C. elegans* ADF: that is, additional uncharacterized TFs might be acting together with LAG-1 to mediate its Notch-independent activating role in the ADF neuron. Thus, CSL TFs display a phylogenetically conserved Notch-independent role in activating neuronal terminal differentiation. We additionally found that LAG-1 is continuously required for neuron cell fate maintenance. Intriguingly, although postembryonic roles of Rbpj have not been studied, some neuronal populations in the adult spinal cord maintain Rbpj expression [50], suggesting that, similar to LAG-1, mammalian RBPJ may display additional Notch-independent roles in cell fate maintenance or neuronal plasticity.

## Material and methods

### Strains

*C. elegans* strains were maintained as described [51]. Wild-type animals are Bristol strain N2. Strains used in this study are listed in S2 Data. Deletion alleles were genotyped by PCR and point mutations by sequencing. Primers are included in S3 Data.

### Generation of *C. elegans* transgenic lines

Gene constructs for *cis*-regulatory analyses were generated by cloning into the pPD95.75 vector. Synthetic construct *tph-1prom46* (GenScript Biotech, Netherlands) and *tph-1prom45* construct were generated by annealing the phosphorylated primers and cloned into pPD95.75. Primers are listed in S3 Data (the whole sequences of these promoters are also included). MEME suite tool [12] and TOMTOM [52] were used for motif enrichment analysis. RTGGGAA [13] and YGTGGGAA [53] LAG-1 consensus sites where used for the identification of additional sites. Site-directed mutagenesis was performed by Quickchange II XL site-directed mutagenesis kit (Agilent/Stratagene, Santa Clara, CA, United States). Ectopic *lag-1* expression, ADF *lag-1* rescue experiments, microinjections, and CRISPR/Cas9-mediated strain generation details are included in S4 Data. All primers are listed in S3 Data.

### RNAi feeding experiments and cell type–specific RNAi

RNAi feeding experiments were performed following standard protocols [54]; see S4 Data for more details. PCR fusion of *srh-142* promoter to sense or antisense *lag-1* cDNA sequences were injected for cell-specific *lag-1* RNAi experiments as previously described [55], 100 ng/μl each sense or antisense PCR together with *ttx-3::mcherry* (25 ng/μl) and *rol-6(su1006)* (25 ng/μl) coinjection markers. Primers are listed in S3 Data.

### 5HT staining

*C. elegans* 5HT antibody staining was performed using the tube fixation protocol [56]; exact protocol is described in S4 Data.

## Behavioural and physiological assays

Dauer induction was performed using filtered liquid culture of worms grown at 7 worms/μl for 4 days. Briefly, 300 μl of pheromone containing extracts or control extracts (culture media alone) were added to 60 mm of IPTG RNAi-seeded plates. After drying, gravid worms were added within a drop of alkaline hypochlorite solution and maintained at 27 ˚C. The percentage of dauer animals was scored in each condition, 72 hours later (P0). For *tph-1* reporter quantification at dauer, the intensity level of the fluorescent reporter was quantified in the ADF neuron ($n > 60$ neurons). For pharyngeal pumping rate, quantification of a minimum of 12 young adult worms fed on *lag-1* RNAi (P0) or control bacteria were recorded with an AxioCam MRm monochrome digital camera using a ZEISS Axio Zoom.V16 microscope (Oberkochen, Germany). Pharyngeal pumping rate was measured by counting the contraction of the pharyngeal bulb over a 10-second period. For quantification of lipid accumulation, oil red O staining was conducted as previously reported [57], with slight modifications included in S4 Data. The analysis of the enhanced slowing response was performed as described [25]. Young adult worms fed with *lag-1* RNAi (P0) or L4440 control bacteria were washed with M9 1X buffer for 5 times and transferred for freely crawl (2 hours) to unseeded IPTG plates, whose agar perimeter was delimited with a 4 M fructose (D-Fructose (Merck Millipore, Darmstadt, Germany), 1% Red Congo dye/ddH2O) ring to avoid worms to scape. After that period of fasting, animals were recovered washing the plate with M9 1X medium. To analyze locomotion of worms approaching a source of food, worms were placed in IPTG plates seeded with 25 μL of bacteria 5X concentrated. Animals were recorded, at room temperature (22 ˚C), with a multi-worm tracker device. Four groups of 8 to 15 worms were recorded per RNAi clone, approximately 50 worms per condition. RNAi empty vector, L4440, was considered the control group. Recording and analysis details are included in S4 Data. For calcium imaging recordings sensitised RNAi strain, *rrf-3(pk1426)*, animals carrying a genetically engineered protein, GCaMP3, expressed only in the ADF were used [58]. Young adult animals fed with RNAi from hatching (P0) were washed with M9 1X and treated for 30 minutes with 1 mM tetramisole hydrochloride (Sigma/Aldrich/Merck, Darmstadt, Germany) to paralyse their movements. Worms were transferred inside a methacrylate microfluidics chamber (self-custom at Worcester Polytechnic Institute, Massachusetts, USA) and settled on an inverted fluorescence microscope (Axio VertA.1, Zeiss) connected to a liquid flow system. M13 buffer (30 mM Tris, 100 mM NaCl, and 10 mM KCl) was used as basal solution, and stimulation solution was prepared adding CuSO4 (10 mM) to M13 buffer and adjusting to pH 4. Both buffers contained tetramisole (1 mM) to avoid worm movement. Images were captured with a CCD camera (Prosilica GC2450, Allied Vision Technologies, Stadtroda, Germany) with 20X objective at 2 frames per second with Micro-Manager software [59]. More details of the analysis are included in S4 Data.

## Temperature shift experiment

*glp-1(e2144)* gravid hermaphrodites were placed on prewarmed plates at the restrictive temperature of 25 ˚C. Mothers were removed after 6 hours, and L1 and young adult worms were scored.

## Heat shock treatment

Plates with worms incubated at 20 ˚C were shifted to 37.5 ˚C for 90 minutes and returned to 20 ˚C for 6 hours. *lag-1(om13)* worms were incubated at 15 ˚C pre- and post-heat shock treatment. Fluorescent reporter expression was quantified in the ADF neurons at L1 larvae stage. Embryonic heat shock for ectopic *lag-1* expression and 5HT fate induction are explained in S4 Data.

## Scorings

Scoring was performed using 40X or 63X objective in a Zeiss Axioplan 2 microscope. Micrographs for figures were obtained with a TCS-SP8 Leica Microsystems confocal microscope using 63X objective. Scoring values and conditions are detailed in S1 Data.

For *cis*-analysis and RNAi screening experiments, a minimum of 30 young adult animals per line or per RNAi clone and replicate were analysed; for mutants analysis, 50 individuals were scored (100 neurons). ON, FAINT, or OFF determinations were established qualitatively by direct observation under the fluorescent microscope: lack of detectable GFP signal was considered OFF, and, if GFP expression was clearly weaker than wild type but still detectable, "FAINT" category was included.

For site-directed mutagenesis, mean expression value of wild type lines is considered the reference value. Mutated constructs are considered to show normal expression when GFP expression was 100% to 80% of average expression of the corresponding wild type construct ("+" phenotype). Compared to wild type reference value, 80% to 20% reduction was considered a partial phenotype "+/−". GFP expression detected in less than 20% of the animals was considered as loss of expression ("−" phenotype).

*Cis*-regulatory and mutant analyses were performed at 25 ˚C, while RNAi experiments and intensity level analyses were performed at 20 ˚C, unless other temperature is indicated in S1 Data.

For quantification of intensity levels, measurements of a minimum of 30 worms (60 neurons) were analysed. Intensity levels were evaluated by capturing images of individual ADF neurons at fixed exposure times for each reporter in a Zeiss Axioplan 2 microscope (see S1 Data for exposure times). The external contour of each ADF neuron was delineated, and fluorescence intensity within the entire neuron was quantified using the ImageJ software and normalised by the area of the neuron. All genetic backgrounds or conditions of each experiment were done the same day. Young adult animals were scored, unless other stage is indicated.

## Statistics

Mutant and RNAi analysis data were categorically classified as "on" or "off," and the significance of the association was examined using the two-tailed Fisher exact test; "faint" phenotype was included as "on" for the statistics. For intensity level of the fluorescent reporters, as samples did not follow a normal distribution, the nonparametric Mann–Whitney test was used with Bonferroni correction for multiple comparisons. For deceleration rate, normality of the data was confirmed and data sets were compared using two-tailed $t$ test. For calcium imaging analysis, the percentage of changes in fluorescence intensity relative to the initial intensity F0, $\Delta F = (F - F0) / F0 \times 100\%$, were plotted as a function of time. The following statistical nomenclature was used: n.s, nonsignificant; $^*p < 0.05$; $^{**}p < 0.01$; $^{***}p < 0.001$.

## Supporting information

**S1 Fig. Primary data for CSL/LAG-1 binding site mutational analysis of ADF serotonin pathway gene CRMs.** CRM, *cis*-regulatory module; CSL, CBF-1, Su (H), LAG-1.
(PDF)

**S2 Fig. *lag-1* mutants show expression defects in ADF serotonergic neuron and RIH and AIM serotonin reuptaker neurons.** n.s, nonsignificant.
(PDF)

**S3 Fig. LAG-1 fosmid reporter expression is similar to the endogenously tagged *lag-1* reporter.**
(PDF)

**S4 Fig. LIM-4 activates *lag-1* expression in the ADF, and LAG-1 is required for cell fate maintenance acting through the minimal ADF CRMs.** CRM, *cis*-regulatory module; n.s, nonsignificant.
(PDF)

**S5 Fig. LIN-12 and GLP-1 Notch receptors are not expressed in ADF neuron terminal lineage.**
(PDF)

**S6 Fig. De novo motif analysis of functional CSL binding site flanking sequences.** CSL, CBF-1, Su (H), LAG-1; WT, wild type.
(PDF)

**S7 Fig. LIN-12 and GLP-1 Notch receptors are not required for *tph-1* up-regulation under specific stimuli.** L1, first larval stage; YA, young adult.
(PDF)

**S8 Fig. DAF-19 does not regulate LAG-1 expression, and additional CSL sites are involved in *tph-1* response upon bacterial infection.** CSL, CBF-1, Su (H), LAG-1; n.s, nonsignificant.
(PDF)

**S1 Data. Raw numerical values.**
(XLSX)

**S2 Data. Strains.**
(XLSX)

**S3 Data. Primers.**
(PDF)

**S4 Data. Supporting methods.**
(PDF)

## Acknowledgments

We thank CGC (P40 OD010440) for providing strains; Dr Laura Chirivella, Noemi Daroqui, and Elia García for technical help; Dr Carlos Mora for building *tph-1* and *mod-5* endogenous reporters; Dr Rafal Ciosk for sharing *glp-1* endogenous reporter; and Dr Iva Greenwald, Dr Luisa Cochella, and Dr Ines Carrera for scientific discussion and paper comments.

## Author Contributions

**Conceptualization:** Miren Maicas, Ángela Jimeno-Martín, Mark J. Alkema, Nuria Flames.

**Data curation:** Miren Maicas, Ángela Jimeno-Martín.

**Formal analysis:** Miren Maicas, Ángela Jimeno-Martín.

**Funding acquisition:** Nuria Flames.

**Investigation:** Miren Maicas, Ángela Jimeno-Martín, Andrea Millán-Trejo.

**Methodology:** Miren Maicas, Ángela Jimeno-Martín, Andrea Millán-Trejo, Mark J. Alkema, Nuria Flames.

**Project administration:** Nuria Flames.

**Resources:** Nuria Flames.

**Supervision:** Mark J. Alkema, Nuria Flames.

**Validation:** Miren Maicas, Ángela Jimeno-Martín, Andrea Millán-Trejo.

**Visualization:** Miren Maicas, Nuria Flames.

**Writing – original draft:** Nuria Flames.

**Writing – review & editing:** Miren Maicas, Ángela Jimeno-Martín, Andrea Millán-Trejo, Mark J. Alkema, Nuria Flames.

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
