## [Editor Report · Decision Letter 0]

13 Feb 2021

Dear Dr Flames, 

Thank you for submitting your manuscript entitled "Notch independent functions of LAG-1/CSL control terminal differentiation, fate maintenance and gene expression plasticity of a chemosensory neuron" for consideration as a Research Article by PLOS Biology. Please accept my apologies for the delay incurred while we sought external advice.

Your manuscript has now been evaluated by the PLOS Biology editorial staff, as well as by an academic editor with relevant expertise, and I'm writing to let you know that we would like to send your submission out for external peer review.

Please re-submit your manuscript within two working days, i.e. by Feb 16 2021 11:59PM.

Kind regards,

Roli Roberts

Senior Editor

PLOS Biology

---

## [Decision Letter · Decision Letter 1]

19 Mar 2021

Dear Dr Flames,

Thank you very much for submitting your manuscript "Notch independent functions of LAG-1/CSL control terminal differentiation, fate maintenance and gene expression plasticity of a chemosensory neuron" for consideration as a Research Article at PLOS Biology. Your manuscript has been evaluated by the PLOS Biology editors, an Academic Editor with relevant expertise, and by three independent reviewers.

You'll see that all three reviewers are broadly positive about your study. However, they do have some requests, some involving additional experimental work (especially the three major points from reviewer #1), that should be addressed before further consideration.

In light of the reviews (below), we will not be able to accept the current version of the manuscript, but we would welcome re-submission of a much-revised version that takes into account the reviewers' comments. We cannot make any decision about publication until we have seen the revised manuscript and your response to the reviewers' comments. Your revised manuscript is also likely to be sent for further evaluation by the reviewers.

We expect to receive your revised manuscript within 3 months. 

**IMPORTANT - SUBMITTING YOUR REVISION**

*Re-submission Checklist*

*Published Peer Review*

*PLOS Data Policy*

*Blot and Gel Data Policy*

Sincerely,

Roli Roberts

Senior Editor,

rroberts@plos.org,

PLOS Biology

REVIEWERS' COMMENTS:

Reviewer #1

[see attachment for formatted version]

This manuscript by Miren Maicas and colleagues describes a novel function of the transcription factor LAG-1 in terminal differentiation of a pair of ADF serotonergic chemosensory neurons in the nematode C. elegans. Using sequence motif analyses, the authors identified a consensus LAG-1-binding motif in promoter regions of genes essential for producing serotonergic phenotypes, including 5-HT synthesis enzyme tryptophan hydroxylase/tph-1, vesicular monoamine transporter/VMAT/cat-1 and 5-HT uptake transporter/SERT/mod-5. The authors generated a series of GFP reporters of the promoter elements, and demonstrated that this LAG-1 binding motif is necessary for baseline expression of these serotonergic phenotype genes in the ADF neurons, but not in any other serotonergic neurons. Using available loss-of-function lag-1 alleles and RNAi knockdown of lag-1 in larval worms, the authors showed that LAG-1 is essential for the expression of these serotonergic phenotype genes in ADF. They further demonstrated that lag-1 deficiency leads to a wide range of physiological and behavioral deficits previously shown to be regulated by ADF 5-HT signaling. Based on these data, the authors concluded that LAG-1 serves as a terminal selector to induce and maintain the terminal differentiation program of the ADF serotonergic neurons.

 An interesting twist from this study is that LAG-1, which was previous known as an effector of the LIN-12/GLP-1 Notch signaling, regulates ADF gene expression independently of LIN-12/GLP-1. Instead, they found that LAG-1 expression in the ADF neurons requires the LIM-domain homeobox gene LIM-4. 

The experiments are well done and data are solid. For examples, LAG-1 effects were characterized in two genetic lag-1 alleles and further validated by lag-1 RNAi, and authors characterized temperature-sensitive mutants to exclude potential maternal LIN-12/GLP-1 contributions. 

This work is significant. Their findings are consistent with emerging evidence indicating that distinct serotonergic neurons are regulated by distinct intrinsic mechanisms and external cues to influence distinct aspects of physiological processes and behavior in mammals. Notch signaling-independent LAG-1 function is novel, showing that transcription factors involved in early developmental processes may be repurposed to control neuron-specific phenotypes in terminal differentiation. 

 I have following three recommendations for revision:

1. Given that LAG-1 is expressed in many neurons, the authors should validate the impact of lag-1 deficiency on ADF function by rescuing ADF phenotypes observed in lag-1 (om13) or lag-1 RNAi worms. This is a common approach in C. elegans research, and should be done. 

2. While their results clearly showed requirement of LAG-1 for baseline expression of serotonergic phenotype genes in ADF, to conclude that LAG-1 is sufficient to induce 5-HT phenotypes, the authors need to demonstrate that ectopically expression of LAG-1 can trigger serotonergic phenotypes in other neuronal types. Alternatively, the authors may consider to revise the conclusion. 

3. In the abstract and text, the authors claimed LAG-1 orchestrating “the protracted actions of terminal differentiation and cell fate maintenance” of ADF. However, there is no data indicating that ADF cell fate is changed in lag-1 mutant worms. Contrary, ADF can upregulate tph-1 expression in response to a variety of neuronal and environmental cues in lag-1 mutants or following lag-1 RNAi (Figures 6 and 7), indicating that the general ADF cell fates and ability to produce 5-HT were preserved. The authors should provide additional data to support their claim, or revise the conclusion. 

Reviewer #2:

This manuscript by Maicas et al. provides strong evidence that LAG-1/CSL. best known for its role as the DNA binding component of the Notch nuclear complex, is required to induce and maintain expression of genes in the ADF serotonergic neuron; the target genes include not only serotonin pathway genes but also other genes involved in ADF-mediated behaviors and genes that are differentially regulated in response to environmental stimuli. Unusually for CSL, this is independent of Notch, a conclusion that is supported by multiple lines of evidence. The work is excellent, and in my opinion, up to the standards of PLoS Biology; I find the main conclusions convincing, and I believe it is of interest from both a neuroscience and a Notch perspective. 

Specific comments:

(1) Avoid priority statements like "This is the first report in C. elegans of a transiently activated signal-regulated TF re-purposed to orchestrate the protracted actions of terminal differentiation and cell fate maintenance" and l. 75 " which has not been reported before in C. elegans". 

(2) Abstract: I think it would be worth mentioning that ADF is one of three serotonergic neuron types, which share serotonergic gene expression but not other effector genes, and LAG-1 is the terminal selector for ADF but not the others. 

(3) lines 127-128: perhaps I missed it, but the synthetic construct contains three copies of the 40bp minimal region that contains the LAG-1 binding site for ADF expression. What happens if fewer modules are present, and what is their relative configuration in the construct shown (all in the same direction?)? If the information is known but not in the paper, please provide it for aficionados.

(4) line 148. Please refer the reader to the figure showing the alteration in the lag-1(q385) null allele when you first mention it. However, at first glance seeing that it is a relatively late stop may give some readers pause (although an effect on ADF target genes is later shown by RNAi so not a major issue--and could be mentioned here as "see below"). If another lag-1 null allele was examined for, say, tph-1, or if rescue was assessed, it should be mentioned. 

(5) lines 213-214. When postembryonic knock down of lag-1 by feeding rrf-3(pk1426) animals was performed, were postembryonic phenotypes expected of lag-1 loss of function in Notch mediated processes also seen?

(6) One point that I think should be clarified relates to imaging conditions and conclusions about gene expression. I could not find any description of the illumination intensity or duration; the scoring will completely depend on this, and if the same conditions/criteria were used in all experiments. Similarly, there is no real description of the distinctions between "on," "faint," and "off". Was there difference in scoring in Figs. 2 and 3 since only 2 has the 'faint" category? What criteria were used in the supplemental figures? Some of the effects seem weaker when just "on" and "off" are used. 

(7) Figure 2. I suggest that you label the "LAG-1 domain" as "DNA binding domain" (we know it's lag-1 but don't know what domain is being featured). Also, I suggest making absence of expression a white bar so it is more clearly distinguished from the others, and perhaps leading with the two endogenous reporters, and then having the transgene reporters follow. Endogenous genes are better than transgenes because of copy number and other potential effects, and it's great to have two endogenous genes in this analysis. Finally, I thought it potentially interesting that loss of lag-1 seems to be a stronger effect on the endogenous targets than transgene reporters. Is that worth a comment? Are all the transgenes made the same way (same markers, same concentrations, etc.)?

(8) Figure 3. the ADF branch practically disappears in pale yellow. Please consider using a color scale that highlights this neuron.

(9) Supp. Fig. 4: The large lineage is almost impossible to navigate. Consider showing only the relevant branches: the ADF branch and the branches leading to other serotonin neurons for which LAG-1 is not expressed; and the branches for RIB, AIM and RIH. 

(10) two typos I noticed:

line 75 typo, Homeodomain.

Supp Fig. 4: capitalize Notch.

Reviewer #3:

In this manuscript, Maicas et al. analyzed the mechanisms that regulate the terminal differentiation of the C. elegans serotonergic neuron ADF. Using cis-regulatory region analysis and loss-of-function experiments, they found that the CSL transcription factor LAG-1 acts as a terminal selector for the ADF neuron. This is a surprising result as CSL transcription factors are mostly known as mediators of the Notch pathway that turns them from repressors to activators. In addition, the authors provide data indicating that the role of LAG-1 as terminal selector is independent of the Notch pathway, although they did not identify the mechanism allowing LAG-1 to activate transcription in the absence of Notch signaling. To sum up, this is a very interesting study that deserves publication.

Overall, this is a careful study and the manuscript is clearly written. However, a few points could be addressed before publication.

- The authors suggest that LAG-1 may cooperate with another transcription factor that will turn it to an activator independently of Notch. One way to identify such factor could be to search for the presence of a conserved binding site close to the LAG-1 binding site in the cis-regulatory regions of ADF effector genes. It would nice to provide the sequence alignments between different Caenorhabditis species for each of the minimal ADF cis-regulatory regions that they have identified. This may reveal the presence of conserved putative binding sites close to LAG-1 binding sites. Related to this, it would be interesting to know what are the other motifs (in addition to the CSL motif) identified during their motif enrichment analysis.

- p8, line 169: "thermosensitive hypomorphic allele lag-1(om13)". The authors could tell us a bit more about this allele. What are the restrictive and the permissive temperatures ? Did they try temperature shifts at different time points ?

- p9, line 201: The data with the fosmid reporter LAG-1::GFP (vlcEx496) are not provided. The authors should show them.

- p16, line 359 and p17, line 381 : This is Figure S6 not S5.

- p17, line 381-382: The authors suggest that the response to pathogenic bacteria could be mediated via an additional LAG-1 binding site. Have they tried to mutate this site ?

- Fig. 1: panel G shows two CSL sites in cat-1prom14 while panel C shows only one.

- Fig. 4D, 4F, 4G: are these curves from a representative single neuron or mean curves from several neurons ?

- Sup. Fig. 6: the authors should explain why a daf-12(sa204) is present in the background.

---

## [Editor Report · Decision Letter 2]

16 Jun 2021

Dear Dr Flames,

Thank you for submitting your revised Research Article entitled "Notch independent functions of LAG-1/CSL control terminal differentiation, fate maintenance and gene expression plasticity of a chemosensory neuron" for publication in PLOS Biology. The Academic Editor has kindly assessed your revisions and responses to reviewers, thereby saving us another round of review. 

Based on this assessment, we will probably accept this manuscript for publication, provided you satisfactorily address the following data and other policy-related requests.

IMPORTANT:

a) Please could you change the title to something slightly more informative? We suggest "The transcription factor LAG-1/CSL plays a Notch-independent role in controlling terminal differentiation and fate maintenance of serotonergic chemosensory neurons"

b) Please address my Data Policy requests below; specifically, please supply numerical values underlying Figs 2BC, 3C, 4AB, 5ABCDEFG, 6BC, 7ABCDFEGH, 8BCDEFG, S2ABCD, S4ABC, S7AB, S8AD and cite the location of the data clearly in each relevant main and supplementary Fig legend.

c) Please try to make your Abstract more accessible, trying to reduce jargon and spelling out abbreviations.

We expect to receive your revised manuscript within two weeks. 

*Published Peer Review History*

*Early Version*

Sincerely,

Roli Roberts

Senior Editor,

rroberts@plos.org,

PLOS Biology

DATA POLICY:

Regardless of the method selected, please ensure that you provide the individual numerical values that underlie the summary data displayed in the following figure panels as they are essential for readers to assess your analysis and to reproduce it: Figs 2BC, 3C, 4AB, 5ABCDEFG, 6BC, 7ABCDFEGH, 8BCDEFG, S2ABCD, S4ABC, S7AB, S8AD. NOTE: the numerical data provided should include all replicates AND the way in which the plotted mean and errors were derived (it should not present only the mean/average values).

We require the original, uncropped and minimally adjusted images supporting all blot and gel results reported in an article's figures or Supporting Information files. We will require these files before a manuscript can be accepted so please prepare and upload them now. Please carefully read our guidelines for how to prepare and upload this data: https://journals.plos.org/plosbiology/s/figures#loc-blot-and-gel-reporting-requirements 

DATA NOT SHOWN?

---

## [Editor Report · Decision Letter 3]

21 Jun 2021

Dear Dr Flames,

On behalf of my colleagues and the Academic Editor, Bing Ye, I'm pleased to say that we can in principle offer to publish your Research Article "The transcription factor LAG-1/CSL plays a Notch-independent role in controlling terminal differentiation, fate maintenance and plasticity of serotonergic chemosensory neurons" in PLOS Biology, provided you address any remaining formatting and reporting issues. These will be detailed in an email that will follow this letter and that you will usually receive within 2-3 business days, during which time no action is required from you. Please note that we will not be able to formally accept your manuscript and schedule it for publication until you have made the required changes.

PRESS: We frequently collaborate with press offices. If your institution or institutions have a press office, please notify them about your upcoming paper at this point, to enable them to help maximise its impact. If the press office is planning to promote your findings, we would be grateful if they could coordinate with biologypress@plos.org. If you have not yet opted out of the early version process, we ask that you notify us immediately of any press plans so that we may do so on your behalf.

Sincerely,

Roli Roberts

Roland G Roberts, PhD 

Senior Editor 

PLOS Biology

rroberts@plos.org